# An In-Networking Double-Layered Data Reduction for Internet of Things (IoT)

**DOI:** 10.3390/s19040795

**Published:** 2019-02-15

**Authors:** Waleed M. Ismael, Mingsheng Gao, Asma A. Al-Shargabi, Ammar Zahary

**Affiliations:** 1College of Internet of Things (IoT) Engineering, Hohai University, Changzhou Campus, Changzhou 213022, China; waleed.m@hhu.edu.cn; 2Faculty of Computer Science, University of Science and Technology, Sana’a 31220, Yemen; a.alshargabi@gmail.com; 3Faculty of Computing and IT, Sana’a University, Sana’a 31220, Yemen; aalzahary@gmail.com

**Keywords:** data fusion, data filtering, Kalman filter, data reduction

## Abstract

Due to the ever-increasing number and diversity of data sources, and the continuous flow of data that are inevitably redundant and unused to the cloud, the Internet of Things (IoT) brings several problems including network bandwidth, the consumption of network energy, cloud storage, especially for paid volume, and I/O throughput as well as handling huge amount of stored data in the cloud. These call for data pre-processing at the network edge before data transmission over the network takes place. Data reduction is a method for mitigating such problems. Most state-of-the-art data reduction approaches employ a single tier, such as gateways, or two tiers, such gateways and the cloud data center or sensor nodes and base station. In this paper, an approach for IoT data reduction is proposed using in-networking data filtering and fusion. The proposed approach consists of two layers that can be adapted at either a single tier or two tiers. The first layer of the proposed approach is the data filtering layer that is based on two techniques, namely data change detection and the deviation of real observations from their estimated values. The second layer is the data fusion layer. It is based on a minimum square error criterion and fuses the data of the same time domain for specific sensors deployed in a specific area. The proposed approach was implemented using Python and the evaluation of the approach was conducted based on a real-world dataset. The obtained results demonstrate that the proposed approach is efficient in terms of data reduction in comparison with Least Mean Squares filter and Papageorgiou’s (CLONE) method.

## 1. Introduction

Over the last few decades, the world has witnessed a dramatic increase in the number of people who utilize Internet services. Accordingly, many propositions are made to use the Internet to connect the items of our daily life for communication purposes [1]. In conjunction with the remarkable advances in wireless sensor technologies and the introduction of small, inexpensive sensors, the items of our daily [2] life have become able to interconnect with each other for data exchange or connect to the Internet to make data available for access. An ever-increasing number of Internet-enabled devices form what is called the Internet of Things (IoT) [3]. These devices facilitate for applications the processing of a massive volume of heterogeneous data, including scalar data (e.g., temperature) and multi-media data (e.g., images), with high speed. This has given rise to many considerable network and cloud challenges [4].

In fact, Internet-enabled devices send continuously flowing data through network-edge gateways and routers that propagate the received data through networks to the backend systems such as cloud databases [5]. However, the ever-growing number of devices produce and send many redundant [6,7] and unused data. For instance, temperature degree does not change frequently [8]. Suppose that temperature and relative humidity data are collected from four sensors deployed in an office and each sensor generates 2 bytes every second. In 1 h, 28.125 Kb would be reported and, in one day, 675 Kb would be reported. This total would escalate exponentially as sensors increase in number and type. Accordingly, the current network infrastructure might not be able to handle such overwhelming data [9], which may result in a high burden on network bandwidth, and I/O throughput [5,10] as well as the difficulty of handling a huge amount of data on the backend systems [7,11,12].

Moreover, network energy consumption gets highly stressed by the transmission of a huge amount of redundant and unnecessary IoT data. This issue is aggravated as the number of IoT devices is increased, especially if there are many network-edge devices dedicated to handling heavy traffic [5]. Cloud storage space is another issue for storing redundant and unnecessary data, especially for pre-paid volume. Therefore, these problems have given rise to data reduction [13] and pre-processing directly close to data sources. Network-edge-based data reduction not only leads to saving storage space, but also reduces the burden on network energy consumption, network bandwidth and I/O throughput, and makes the usage of available resources more efficient [4].

IoT data are typically time series data [14] that can lead to the problems mentioned above [5]. Many state-of-the-art network-edge-based data reduction approaches have been proposed to address such problems, as described in the literature review below, but they are limited. Therefore, this paper proposes a network-edge-based approach that is based on data filtering and fusion techniques to relieve the aforementioned problems. The data filtering of the proposed approach is based on data change detection for the avoidance of redundant data transmission and based on the deviation (>$e_{\max}$) of real observations from their estimated values calculated by Kalman filter. Therefore, the real observations that have small deviation (≤$le_{\max}$ ) from their estimated ones can be recovered by estimation. We also propose a data fusion method that is based on a minimum mean square error criterion. It employs a simple strategy to fuse data of the same time domain for certain a area.

According to Anastasi et al. [15], data reduction techniques fall into three categories: data compression, data forecasting and in-network processing. In data compression techniques, different data encoding techniques are applied to reduce the size of data and then data are sent in reduced form. In forecasting techniques, the method is maintained on two sides, such as sensor and sink nodes or gateways and the cloud data center. The method forecasts the value and calculates the difference from the actual value. If the difference is within the predetermined error range, there is no need for sending the actual value to the other side. In network processing techniques, the data are processed by an intermediate device, such as gateway/router, and then the intermediate device sends only the processed data instead of raw data [16]. The last type of data reduction techniques is the concern of this paper.

The rest of the paper is organized as follows. Firstly, we review some state-of-the-art data reduction techniques. Then, we provide a background about Kalman filter and describe the architecture of the proposed approach. Next, we present the evaluation of the proposed approach and discuss the obtained results. Finally, we sum up with conclusion, along with future work.

## 2. Related Work

Ever-growing IoT devices that produce and send a massive volume of unnecessary and useless data to the cloud lead to the problems of network bottleneck and to the difficulty of handling such voluminous data at the cloud. Network-edge-based data reduction is an option to address such problems. Many state-of-the-art methods employ network edge for reducing data before transmitting it to the cloud to address such problems. In Reference [5], the authors proposed a real data reduction approach to solve the problems of network bandwidth, I/O throughput, network energy consumption and cloud storage. They proposed NECtar that is able to automatically switch between different data reduction algorithms, including sampling, selective forwarding, piecewise approximation, perceptually important points (PIP) algorithm and data change detection, based on the data type. In addition, Feng et al. [4] proposed an approach to solve the energy consumption issues of IoT devices and the depletion of cloud storage. Their approach is based on multi-tier data reduction mechanism that functions on two tiers, gateway and network edge tier. Their approach applies the PIP algorithm on time series data. Moreover, they integrated the PIP algorithm with several techniques, including interval restriction, dynamic caching and weight sequence selection. At the second tier, they introduce data fusion method based on optimal dataset selection to fuse the same-time domain data for a specific location. Other authors introduced an approach that takes advantage of edge redundant hardware to increase the performance of data reduction and to deal with failure of IoT gateways (e.g., see Reference [9]). The authors employed PIP algorithm in their approach for increasing the performance of data reduction and utilized primary and secondary IoT gateway without coordination between them and without sending redundant data. The idea is that the dataset is split into odd and even datasets and processed by primary and secondary IoT gateways, respectively. Although PIP algorithm considers the necessary points for keeping the overall shape of data, it suffers from computational complexity as it needs longer run time to identify the important points that contribute in the overall shape of time series data in each iteration [17] and has a worst case complexity than the method in Reference [18].

Other solutions employ dual filter to perform data reduction at network edge. In Reference [19], the authors proposed an approach that exploits the fog computing for data reduction. The approach is based on two phases. In the first phase, the data are modeled based on multivariate normal distribution. In the second phase, a Kalman filter with the same parameters is deployed in both fog and cloud platform. The estimated values of measured data by end devices are calculated on both sides, cloud and fog platform, based on historical data and internal data correlation. If the predicted values exceed the prediction range, the observations are forwarded to the cloud. In contrast, the authors of [20] proposed adaptive approach for data reduction that is based on least mean square (LMS). Their approach is based on dual LMS filter, one deployed at the source (sensor nodes) and another in the sink node. The sources have to send their measured values that deviate (>user input value) from their predicted values. Both approaches [19,20] rely on caching historical data, which causes computing overhead. However, the proposed approach caches only one measured value for each data source when a change in the data takes place. Our proposed approach employs Kalman filter for data filtering, as described below.

### Kalman Filter

Kalman filter is known as a recursive algorithm for optimal estimating future states of dynamic systems [21]. It is commonly used as an estimation algorithm [22] and it is also known as an adaptive filter [23] to combine previously estimated values with current measurements. One of its advantages is its fast error convergence and better estimation through changing the measurement covariance by controlling fundamental parameters [21]. Consider the sensor readings at time series:$$z_{t}\;=\;\left[z_{t},z_{t-1},...,z_{t-N-1}\right]$$
that is composed from the previous reading from time, and forms the input measured data to the Kalman Filter algorithm with measurement noise according to Equation (Equation 1) [20].
(1)$$z_{t}=Hx_{t}+v_{t}$$
where *H* represents an identity matrix, $v_{t}$ is the uncertainty in the observation and $z_{t}$ represents the observation vector at time *t*. Kalman filter is based on two phases: prediction and update. In the prediction phase, the estimated value $x_{t}$ and covariance matrix $P_{t}^{-}$ are estimated according to Equations (Equation 2) and (Equation 3), respectively.

(2)$$x_{t}\;=\;Ax_{t-1}+Bu_{t-1}+w_{t}$$

The corresponding covariance matrix is calculated as:(3)$$P_{t}^{-}=AP_{t-1}A^{t}+Q$$
where $x_{t}$ is the posteriori state, $x_{t-1}$ is the a priori state, $A^{t}$ is the state transition model matrix, *B* is input control matrix, $u_{t-1}$ is the control vector and *Q* is the white Gaussian noise covariance matrix [24]. In the update phase, Kalman Filter gain is calculated based on Equation (Equation 4), the estimated state $x_{t}$ is updated according to Equation (Equation 5) and the observation noise covariance according to Equation (Equation 6)
(4)$$K_{t}=P_{t}^{-}H^{T}{\left(HP_{k}^{-}H^{T}+R\right)}^{-1}$$
where *R* and $H^{T}$ are the covariance of observation noise and the observation model, respectively. Accordingly, the estimated state ${\hat{x}}_{t}$ and covariance matrix $P_{t}$ are updated based on Kalman gain as:(5)$${\hat{x}}_{t}={\hat{x}}_{t-1}+K_{t}\left(z_{t}-H{\hat{x}}_{t-1}\right)$$
(6)$$P_{t}=\left(I-K_{t}H\right)P^{-}$$
Altogether, both the observation covariance matrix $P_{t}$ and estimation ${\hat{x}}_{t}$ are predicted based on the values of pervious time instance and updated after getting new measured data [21].

## 3. Proposed Approach

In this paper, an in-networking data filtering and fusion mechanism for reducing IoT sensor data is proposed. Our approach employs a network edge for filtering and fusing data before data transmission to the cloud occurs, as shown in Figure 1.

Consider N number of sensors deployed in a specific area to monitor their surroundings such as monitoring the changes of temperature degree, humidity, wind speed, etc. At each time instance t>0, each sensor node $S_{i}$
(i=1,2,3,...,N) produces data streams z(t) about its surrounding and sends it to an intermediate node such as a base station or router/gateway, to propagate such data to the cloud. Data streams are a sequence of data packets in consecutive order [20].

In this paper, we are interested in reducing the amount of IoT data transmitted over the network to mitigate the network bottleneck problems prior to data transmission. Firstly, the proposed approach reduces the sensor reading through removing redundant data. When there is no change in sensor readings detected, there is no need to send the sensor actual reading to the cloud. For instance, in temperature monitoring application, temperature degree does not change frequently. The sensor keeps producing temperature readings that are not changing over a short period of time. Even if there is a change in sensor readings detected, not all data are important. Therefore, by calculating the deviation between the actual reading and its estimated value calculated by Kalman filter, the data are discard or forwarded to the data fusion layer. In other words, the actual sensor reading doe not need to be forwarded to the data fusion layer unless the deviation between the sensor actual reading and its estimated value (>predefined Maximum Absolute Error). According to Figure 1, each sensor reading is passed through filtering component, which is responsible for checking for redundant data and calculating the deviation of the actual sensor reading and its estimated value, and then decides to either forward it to the data fusion layer or discard it. More details are explained in the next sections. As IoT data are typically data series, the data filtering layer receives simultaneous actual readings of sensors. Such data are filtered before passing them to the data fusion layer. In this case, not all data received by the data filtering layer will be passed. Therefore, the data fusion layer aggregates the received data from the data filtering layer to construct an accurate representation of the phenomenon that is under observation. The output of the data fusion layer will be propagated to the cloud. This proposed approach can be applied by both critical and not critical applications through customizing the deviation value.

Figure 1 gives an illustrative representation of the proposed approach. There are three main layers: the data source (a set of N sensor nodes), the data filtering layer dedicating a filtering component for each sensor node and the data fusion layer. The two following sections provide the details of the data filtering layer and the data fusion layer.

### 3.1. Data Filtering Layer

The data filtering layer relies on two techniques. The first technique is called data change detection. According to Coppin and Bauer [25], change detection is defined as “the process of identifying differences in the state of an object or phenomenon by observing it in different times”. In our case, change detection is the process of identifying the difference between the previous and current observation. The second technique is the deviation of the real observations from their estimated values by identifying the maximum absolute error. The idea behind these techniques is to check the current observation z(t) at time *t* with its previously cached observation cached_z in order to detect a data change. This layer receives the first observation z(t) at the first time. Then, it caches it for later use. Afterwards, it compares the next received observation with the cashed value (cached_z). If there is no data change detected, the data are discarded. However, if a data change is detected between the cached value cached_z and z(t) at time *t*, the cached value is updated with current observation z(t) and z(t) is passed through Kalman filter in order to find its estimate x(t). The observed value is passed to the data fusion layer, if there is a significant deviation ($e(t)>e_{\max}$ ) from its obtained estimated value x(t) at time *t*, where $e_{\max}$ is the maximum absolute deviation from the real observation set up by the user and e(t) is calculated according to Equation (Equation 7). The output of this layer is filtered data fd(i)
(i=1,2,3,...,N). The filtered data fd(i) are the input of the data fusion layer:(7)e(t)=|z(t)−x(t)|
where $z_{t}$ is the current observed value and x(t) is its estimated value at time *t*.

Using maximum absolute deviation helps determine and then compare the real observation with its estimated value to find whether the difference between them exceeds the $e_{\max}$ and thus whether to pass the reading to the next step for fusion.

The data filtering layer receives the actual readings of sensor nodes simultaneously at time instance *t* and processes the actual readings of each sensor independently, as shown in Figure 1, through applying the two techniques mentioned above. Some actual readings of sensor nodes received at time instance *t* might be filtered out. Therefore, the data fusion layer is suggested to aggregate the readings that are not filtered out to build an observation that represents the phenomenon under observation.

Another case taken into consideration is faulty sensor reading detection. The proposed approach considers the correlation between sensor readings to find faulty sensor readings. The proposed approach is designed for the purpose of data reduction of sensors deployed in a specific location for a specific time domain. Therefore, if a faulty reading is detected, the faulty reading is replaced by its estimated value calculated by Kalman filter.

The summary of the data filtering algorithm is shown in Algorithm 1, the parameters used are shown in Table 1, and the data flow is represented in Figure 2.

**Algorithm 1** Data filtering operation
**INPUT:** sensor reading z(t)  e_max = user input  **while** true **do**   **if**
z(t) is the first reading **then**    cached_z(i)← zt(i)    Send z(t) to the fuser   **else**    **if**
cashed_z(i) ! = z(t)
**then**     cached_z(i)← z(t)     Call kalman filter to calculate estimated value     e(t)←z(t)−x(t)     **if**
$|e(t)|>e_{\max}$
**then**      fd(i)←z(t)      Send fd(i), R(i) and H(i) to the data fusion layer     **else**      Discard x(t) and z(t) values     **end if**    **end if**   **end if**  **end while**


### 3.2. Data Fusion Layer

The incoming filtered data from the data filtering layer flow continuously to the data fusion layer. Some data from the data filtering layer are missing (filtered out data). Therefore, the purpose of this layer, prior to transmitting data to the cloud, is to fuse data of the same domain for certain location to improve data reliability, remove data redundancy and complement the missing data [4]. A good example of the data fusion is a set of sensors deployed in a data center room in order to monitor the overall temperature.

According to Yukun et al. [21], there are two common Kalman-filter-based methods for the fusion of multi-sensor data. The first method is simple, in which multi-sensor data are simply fused based on the observation vector of Kalman filter. The other method uses a minimum mean square error (MMSE) criterion for the fusion of multi-sensor data [21,24]. The second method is adopted in our proposed approach because it requires low computational load. In this method, the dimension of the observation vector is unchanged and the fused observation is obtained by weighted observation [21].

The main point of this layer is a set of sensors to observe the same phenomenon, such as temperature, humidity, etc. Therefore, the incoming data from the data filtering layer of the same source is fused at the first level, data fusion. The reader is referred to [26] for more information. Suppose that filtered data of *n* sensors $FD=fd_{1},fd_{2},fd_{3},...,fd_{n}$ coming from the data filtering layer at time instance *t* with the covariance matrix $R_{i}$ of each sensor calculated by Kalman filter in the data filtering layer. After receiving the filtered data from the data filtering layer, as shown in Figure 3, the total covariance matrix is calculated based on Equation (Equation 9). Then, the received filtered data are fused and used to obtain the state vector z(t) that will be sent to the cloud. The Kalman filter will be applied on the total state vector z(t) if there are missing data resulting from the data filtering layer and the estimated value will be sent to the cloud instead of z(t). The measurement noise is independent for each sensor. The equation for fusing the measurement vectors $fd_{i}\left(i=1,2,...,N\right)$ at time instance *t* is given by Equation (Equation 8) to obtain the fused measurement vector z(t) [21].
(8)$$z\left(t\right)={\left[\sum_{i=1}^{N}R_{i}^{-1}\left(t\right)\right]}^{-1}\sum_{i=1}^{N}R_{i}^{-1}\left(t\right)fd(i)$$
where $R_{i}$ represents the covariance matrix of measurement vector $z_{i}$. The identity matrix of the fused measurement is given by Equation (Equation 9).
(9)$$H\left(t\right)={\left[\sum_{i=1}^{N}R_{i}^{-1}\left(t\right)\right]}^{-1}\sum_{i=1}^{N}R_{i}^{-1}\left(t\right)H_{i}\left(t\right)$$
where H(t) represent the identity matrix of fused measurement vector z(t) and $H_{i}\left(t\right)$ is the identity matrix of sensor (*i*) at time *t*. In addition, the covariance matrix of the fused measurement vector is calculated according to Equation (Equation 10).
(10)$$R\left(t\right)={\left[\sum_{i=1}^{N}R_{i}^{-1}\left(t\right)\right]}^{-1}$$
where *N* represents the number of sensors. The estimates can be obtained using Kalman Filter as described above.

The summary of the data fusion algorithm is shown in Algorithm 2, the parameters used are shown in Table 2 and the data flow is represented in Figure 3.

**Algorithm 2** Data fusion operation**INPUT:** The output data stream of algorithm_1 FD=(fd1,fd2,...,fdn)Covariance matrix calculated by Kalman filter in data filtering layer R=(R1,R2,...,Rn)Identity Matrix Calculated by Kalman filter in data filtering layer H=(H1,H2,...,Hn)  **if**
fd1=NONEANDfd2=NONEAND...ANDfdn=NONE
**then**  2:  Exit   **else**  4:  R(t)←∑i=1NRi−1(t)−1  5:  z(t)←∑i=1NRi−1(t)−1∑i=1NRi−1(t)fd(i)  6:  H(t)←∑i=1NRi−1(t)−1∑i=1NRi−1(t)Hi(t)    Calculate total estimated value using Kalman filter based on total z(t), R(t) and H(t)    send *z(t)* to cloud.  7:  **if** there is a missing data **then**  8:   send the estimated value calculated by Kalman filter is sent the cloud.  9:  **end if****   end if**


## 4. Implementation and Evaluation of the Proposed Approach

This section presents the implementation and evaluation of our approach. The approach was implemented and executed using Python. As mentioned in Section 3, the proposed approach for IoT data reduction consists of two layers. The first layer is to filter data of each sensor based on data change detection and the deviation of observed value from its Kalman-filter-based estimated value. The second layer is to fuse sensor data of the same resources after filtering to enhance the accuracy of the data since some missing data result from filtering process of the first layer. The performance evaluation of our approach in the first layer and second layer is presented below.

### 4.1. Datasets

In the simulation, real-world datasets from Intel lab were used, as shown in Figure 4. The offered datasets were obtained from 54 Mica2Dot sensors. Each sensor provides data for weather, such as temperature, humidity, light, and voltage, as well as the time at which a sensor reading is acquired. For experiments, we selected datasets (temperature) reported by Sensor 1 (Mote 1), Sensor 2 (Mote 2), and Sensor 3 (Mote 3) between 6 March 2014 and 8 March 2014, resulting in a dataset with 5897 readings, to ensure the efficiency of the proposed approach.

### 4.2. Simulation Environment

A simulation code was written in Python to evaluate our proposed approach. The datasets were from the Intel Lab, and include typical time stamped data of weather such as temperature. The weather data are a time series data for IoT application [4]. We conducted the experiment with the temperature data for Sensors 1–3. Since Kalman filter requires prior information to work properly, we set the first value of sensor reading to obtain the Kalman filter initial state x(0), as shown in Table 3.

To carry out simulation on datasets (temperature), Kalman filter parameters were determined in the first layer (the data filtering layer). We selected the parameter values as follows:$$\begin{array}{c} A=\left[\begin{array}{cc} 1 & 0\\ 0 & 1 \end{array}\right]\\ H=\left[\begin{array}{cc} 1 & 0 \end{array}\right] \end{array}$$
where *Q* is considered as white Gaussian noise and the target noise is *P*. *Q* and *P* were initialized as follows:$$Q=\left[\begin{array}{cc} 2.25e^{-6} & 4.50e^{-6}\\ 4.50e^{-6} & 9.00e^{-6} \end{array}\right]$$
$$P=\left[\begin{array}{cc} 0.000001 & 0\\ 0 & 0 \end{array}\right]$$

In addition, the maximum absolute deviation $e_{\max}$ was set to the range of values $\left(e_{\max}=0.01,0.02,...,0.09\right)$.

## 5. Study Results and Evaluation

The performance of the proposed approach was evaluated based on the results obtained from the two layers as follows.

### 5.1. Data Filtering Layer

As mentioned above, to ensure the efficiency of the proposed approach, it was evaluated based on the readings of three sensors (Sensors 1–3) over the temperature data. To evaluate the proposed approach, we compared the results with two algorithms commonly used in data reduction, namely Least Mean Square (LMS) algorithm and Papageorgius’s method, which employ Perceptually Important Points algorithm (PIP). LMS algorithm was selected as prediction algorithm and the CLONE approach of Papageorgius’s method was selected because it is faster than other two approaches (TWIN and AVG) to avoid the delay and perform a fair comparison. The CLONE approach of Papageorgius’s method was implemented with cache size equal to 100 items. We compared three aspects to evaluate the results of the proposed approach: the number of forwarded items, data reduction percentage of incoming data and data reduction accuracy.

Data reduction percentage is obtained by:
(11)SP=(((AD)−(TD))×100)/100
(12)DRP=100−(SP)
where *SP* is the saving percentage, *AD* represents the actual data, *TD* represents the transmitted data, and *DRP* is the data reduction percentage.Data reduction accuracy is the similarity between actual data and filtered data mixed with estimated data to make a set of the same length as the original one based on the Jaccard coefficient as follows:

If $T1=\left[\left(t_{1}^{1},v_{1}^{1}\right),\left(t_{2}^{1},v_{2}^{1}\right),\ldots ,\left(t_{n}^{1},v_{n}^{1}\right)\right]$ is the real data and $T2=\left[\left(t_{1}^{2},v_{1}^{2}\right),\left(t_{2}^{2},v_{2}^{2}\right),\ldots ,\left(t_{n}^{2},v_{n}^{2}\right)\right]$ is the filtered data, the Jaccard similarity coefficient between them is calculated [4] as:(13)$$\alpha =\frac{\sum_{i=1}^{n}\min(\left(t_{i}^{1},v_{i}^{2}\right))}{\sum_{i=1}^{n}\max(\left(t_{i}^{1},v_{i}^{2}\right))}\times 100$$
where *n* is the length of actual data. A range of maximum absolute error ($e_{\max}$ = 0.01, 0.02,..., 0.09) was used to investigate the impact of $e_{\max}$ on the amount of forwarded data to the data fusion layer, data reduction percentage and data reduction accuracy and to verify and evaluate the results of the proposed approach, LMS algorithm and Papageorgius’s method (CLONE).

Table 4, Table 5 and Table 6 summarize the experiment results of the data filtering layer for Sensors 1–3. They represent the comparison of the proposed approach, LMS and Papageorgious’s method (CLONE) based on the range of Maximum Absolute Error (MAE), as mentioned above. The comparison was performed based on 5896 readings of each sensor. In addition, the number of forwarded data to the data fusion layer is also represented. Moreover, the data reduction percentage and data reduction accuracy were calculated based on Equations (Equation 11)–(Equation 13) for each sensor, respectively. To illustrate the data in Table 4, Table 5 and Table 6, Figure 5, Figure 6 and Figure 7 present the performance of the proposed approach with a comparison to LMS and Papageorgious’s method (CLONE) with $e_{\max}=0.01$ in terms of data filtering. Accordingly, it can be seen that the proposed approach outperformed LMS and Papageorgious’s method (CLONE) for Sensors 1–3. The figures show the temperature data as *y*-axis plotted against the number of samples as *x*-axis.

According to the results in Table 4, it is noted that the proposed approach achieved data reduction percentage in the range 43.25–85.13% with data reduction accuracy in the range 69.41–92.61% in comparison with LMS algorithm, which achieved data reduction percentage in the range 13.61–73.88% with data reduction accuracy in the range 71.92–96.30%. Papageorgious’s method (CLONE) achieved high accuracy in the range 88.89–94.91% with very low data reduction percentage ranging 5.14–9.18%. The results are plotted against $e_{\max}$ in Figure 8. In Table 5, the results show that the proposed approach achieved data reduction percentage in the range 39.52–77.34% and data reduction accuracy in the range 62.60–93.30%, while LMS algorithm achieved data reduction percentage in the range 11.48–75.13% with data reduction accuracy in the range 69.96–71.54%. Papageorgious’s method (CLONE) achieved low data reduction percentage and high data reduction accuracy in the ranges 3.02–5.73% and 97.63–94.32%, respectively. The results are plotted against $e_{\max}$ in Figure 9. Moreover, the results in Table 6 show that the data reduction percentage achieved by the proposed approach is in the range 41.99–84.26% with accuracy in the range 93.17–68.56%, while LMS algorithm achieved data reduction percentage in the range 12.91–73.42% and data reduction accuracy in the range 72.49–96.79%. Papageorgious’s method (CLONE) achieved data reduction percentage in the range 4.40–9.12% with data reduction accuracy in the range 90.92–96.01%. Figure 10 illustrates the results plotted against $e_{\max}$. It can be noted that Papageorgious’s method (CLONE) achieved the highest data reduction accuracy and the lowest data reduction percentage followed by LMS algorithm, which achieved data reduction accuracy higher than the proposed approach but with lower data reduction percentage. Based on the results shown in Table 4, Table 5 and Table 6, we can say that the proposed approach achieved the highest data reduction percentage compared to LMS algorithm and Papageorgious’s method (CLONE). This is attributed to the fact that the proposed approach employs the data change technique to remove the redundant data of each sensor, as described in Section 3.1. Moreover, the proposed approach relies on Kalman filter to calculate the deviation between the sensor actual readings and their estimated values. Kalman filter has an advantage of faster error convergence and better estimation through changing the measurement covariance by controlling fundamental parameters. As $e_{\max}$ increases, the data reduction percentage increases too, while the data reduction accuracy decreases. In fact, as $e_{\max}$ increases, the differences between actual values and their estimated ones increase. Therefore, the data reduction percentage increases, while the data reduction accuracy decreases.

### 5.2. Data Fusion Layer

The proposed data fusion layer employs a minimum-mean-square-error criterion for multisensory data fusion, as described above by Equations (Equation 8)–(Equation 10). For the evaluation of the data fusion layer of the proposed approach, we adapted the data fusion layer of the proposed approach for both LMS and Papageorgious’s method (CLONE) to compare the data reduction percentage of the three approaches relative to the total original readings of Sensors 1–3. The data reduction percentage is calculated according to Equations (Equation 11) and (Equation 12), but relative to the total original readings of Sensors 1–3, instead of each sensor individually. The input of data fusion layer is based on the output of data filtering layer depending on maximum absolute error values ( $e_{\max}$ = 0.01, 0.02,..., 0.09). Moreover, we used the similarity between the final fused data and the original readings of Sensors 1–3 to calculate the data recovery accuracy. As shown in Table 7 and Figure 11, the proposed approach had highest data reduction percentage (69.63–86.32%), while LMS achieved data reduction percentage in the range 66.78–83.51% and Papageorgious’s method (CLONE) achieved data reduction equal to 66.67% and no change was noted because data reduction percentage of data filtering is inconsiderable. Figure 12 represents the performance the data fusion layer with $e_{\max}=0.01$ in comparison with the other methods for which the data fusion layer of the proposed approach is adapted. In the figure, we can see that there is a slight difference in the output of the data fusion layer for each method. This is because of the different filtering techniques used by the methods.

The data recovery accuracy was calculated based on the similarity between the fusion results and the original readings of each sensor using Equation (Equation 13). In the case the data of all sensors were filtered at the same time in the data filtering layer, the data recovery accuracy was calculated by replacing the filtered-out data with the estimated data calculated by Kalman filter. Figure 13 shows that the proposed data fusion layer achieved high similarity between the original readings and filtered readings for different maximum absolute error values ($e_{\max}$ = 0.01, 0.02,..., 0.09). It can be noted that the similarity between the original readings and the filtered readings of Sensor 1 was in the range 89.99–92.94%. The similarity between original readings and the filtered readings of Sensor 2 was the range 80.31–86.98%. The similarity between the original readings and the filtered readings of Sensor 3 was in the range 89.55–92.09%. Figure 13 shows that Sensor 2 had lower similarity than Sensors 1 and 3. This was due to more data filtered in the data filtering layer.

Finally, it is worth noting that the proposed approach achieved the highest data reduction percentage in the data filtering layer with different values of maximum absolute error ($e_{\max}$ = 0.01, 0.02, ..., 0.09) in comparison with LMS method and Papageorgious’s method (CLONE). Moreover, the proposed approach attained acceptable data reduction accuracy, as shown in Figure 13. In the data fusion layer, after adapting the second layer of the proposed approach for LMS algorithm and Papageorgious’s method (CLONE), the proposed approach outperformed LMS algorithm and Papageorgious’s method (CLONE) in terms of data reduction percentage, as shown in Table 4, Table 5 and Table 6, and achieved high data recovery accuracy, as reported in Figure 13.

## 6. Conclusions

In this paper, we propose an approach for IoT data reduction. The proposed approach is an in-networking approach and comprises two layers. The first layer (the data filtering layer) is dedicated for data filtering depending on two techniques, namely data change detection and the deviation (>user input value) of real observations from their estimated values. Moreover, the data filtering layer detects the faulty sensor readings by calculating the correlation between the readings of sensors. The second layer is data fusion layer and it is based on minimum mean square error technique to improve data reliability after filtering operation. Its rule is to fuse the output of the first layer. The essential goal is to reduce data transmitted from network edge to the cloud.

We compared the proposed approach with LMS method and Papageorgious’s method (CLONE). Based on the results, the proposed approach achieved a higher reduction percentage than LMS method and Papageorgious’s method (CLONE), and attained an acceptable data recovery accuracy. Even though the proposed approach has shown good performance and efficiency in filtering and fusing the IoT linear data, future work will focus on investigating the application of approach on nonlinear data and multi-media data as well as the usage of extended Kalman filter. This will also include dealing with multi-rate data. 

References yes

## Figures and Tables

**Figure 1 sensors-19-00795-f001:**
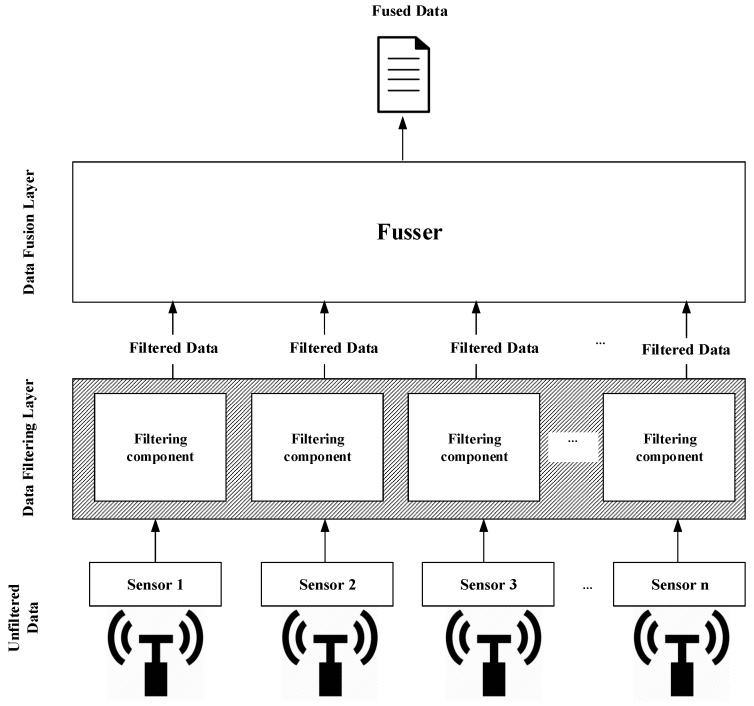
Architecture of the proposed approach.

**Figure 2 sensors-19-00795-f002:**
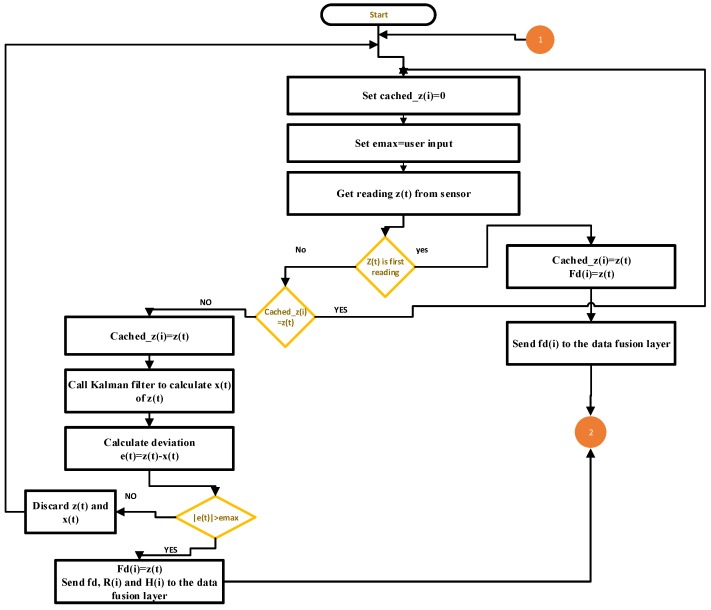
Data flowchart of the data filtering layer.

**Figure 3 sensors-19-00795-f003:**
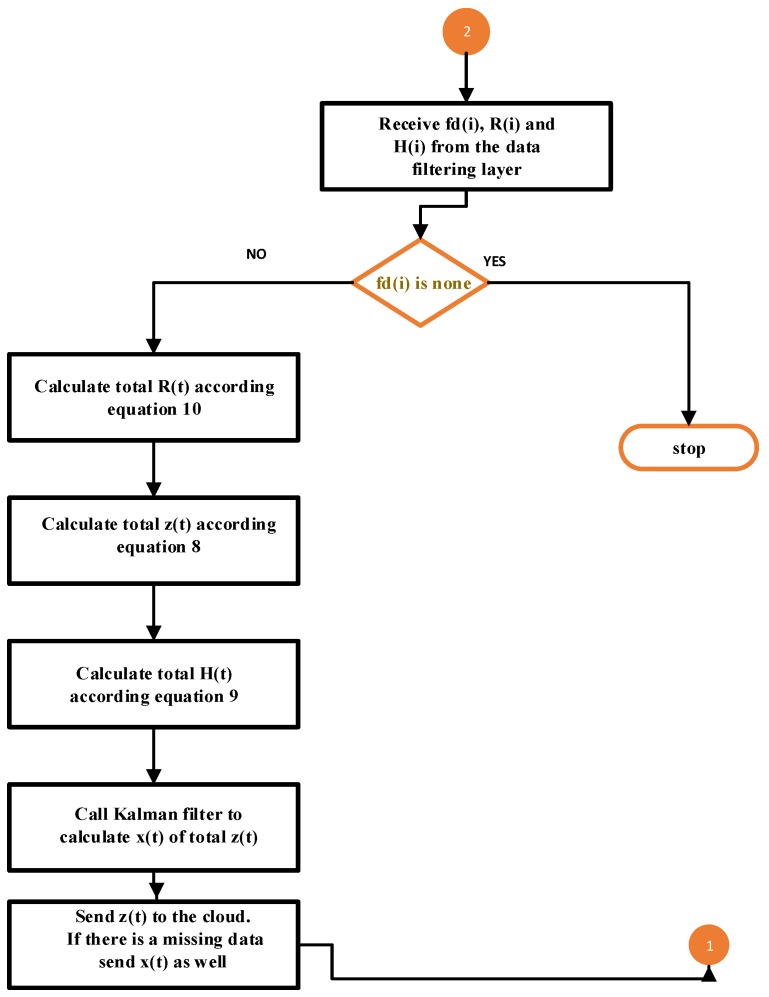
Data flowchart of the data fusion layer.

**Figure 4 sensors-19-00795-f004:**
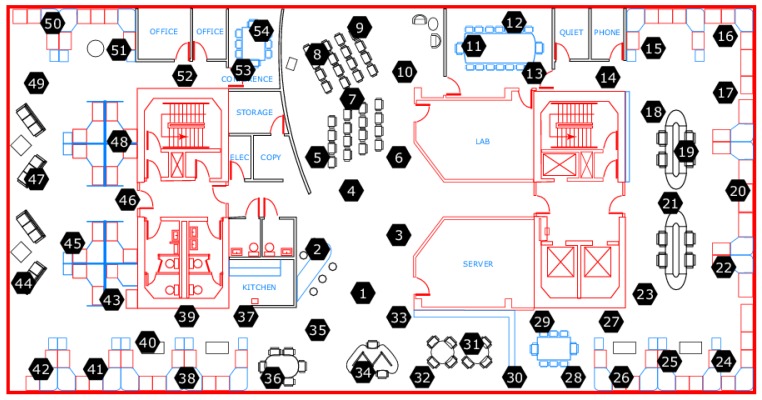
A view for sensors along with weather boards at Intel Berkeley Research lab [27].

**Figure 5 sensors-19-00795-f005:**
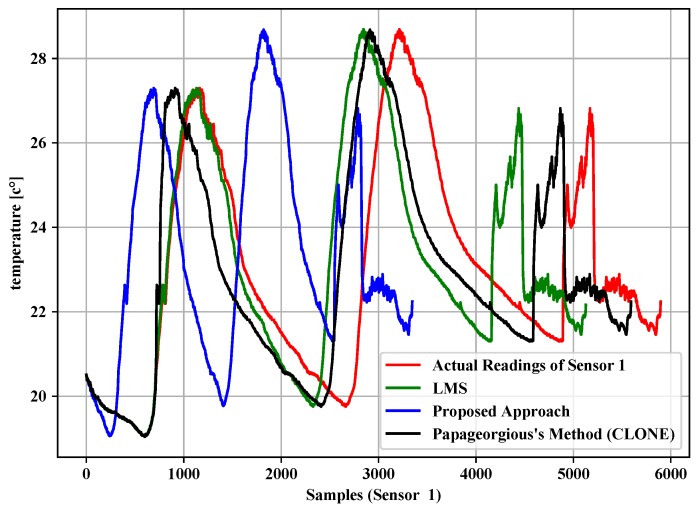
Comparison of the proposed approach, LMS and Papageorgius’s method (CLONE) for Sensor 1 with $e_{\max}=0.01$.

**Figure 6 sensors-19-00795-f006:**
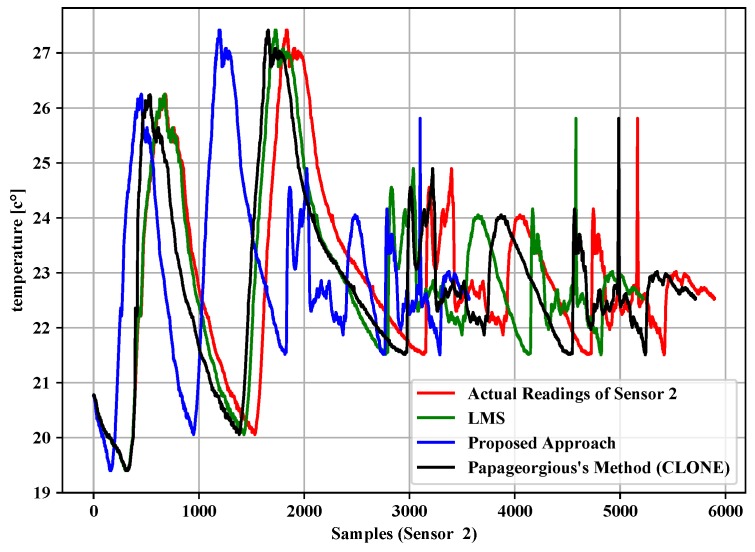
Comparison of the proposed approach, LMS and Papageorgius’s method (CLONE) for Sensor 2 with $e_{\max}=0.01$.

**Figure 7 sensors-19-00795-f007:**
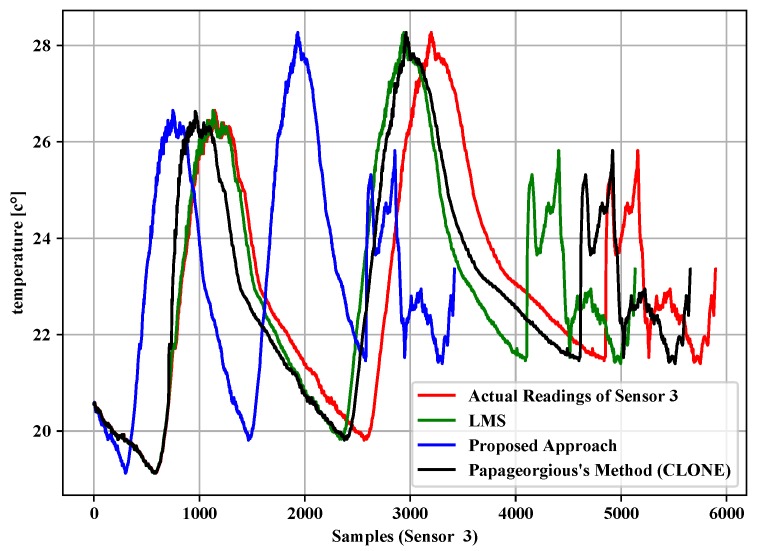
Comparison of the proposed approach, LMS and Papageorgius’s method (CLONE) for Sensor 3 with $e_{\max}=0.01$.

**Figure 8 sensors-19-00795-f008:**
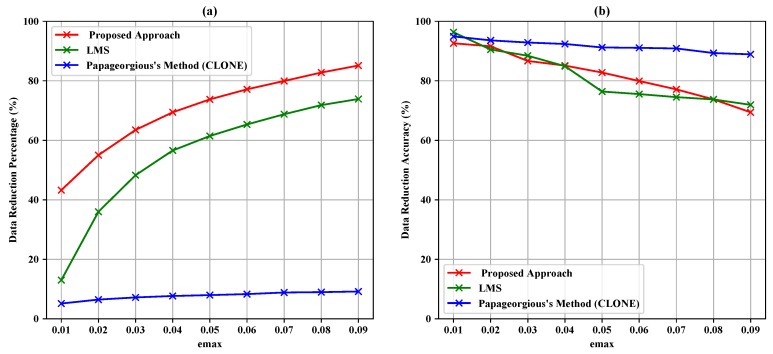
Data reduction percentage (**a**) and data reduction accuracy (**b**) vs. $e_{\max}$ for Sensor 1.

**Figure 9 sensors-19-00795-f009:**
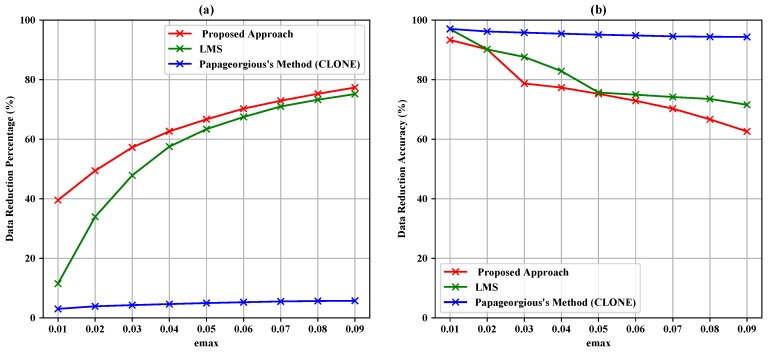
Data reduction percentage (**a**) and data reduction accuracy (**b**) vs. $e_{\max}$ for Sensor 2.

**Figure 10 sensors-19-00795-f010:**
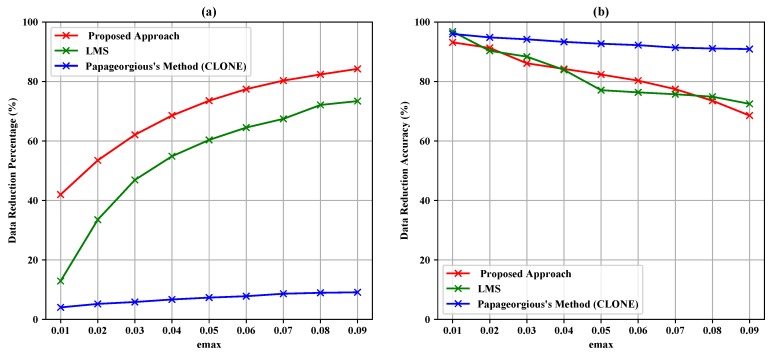
Data reduction percentage (**a**) and data reduction accuracy (**b**) vs. $e_{\max}$ for Sensor 3.

**Figure 11 sensors-19-00795-f011:**
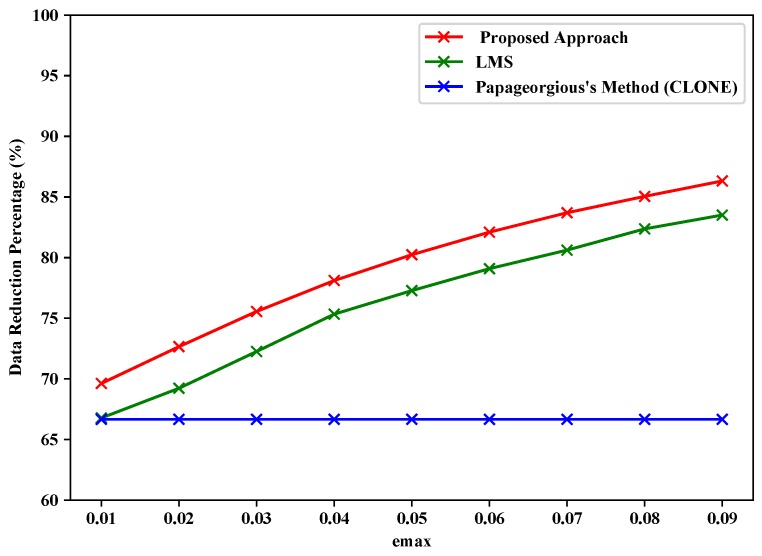
Data reduction percentage vs. $e_{\max}$ for fused data of Sensors 1–3.

**Figure 12 sensors-19-00795-f012:**
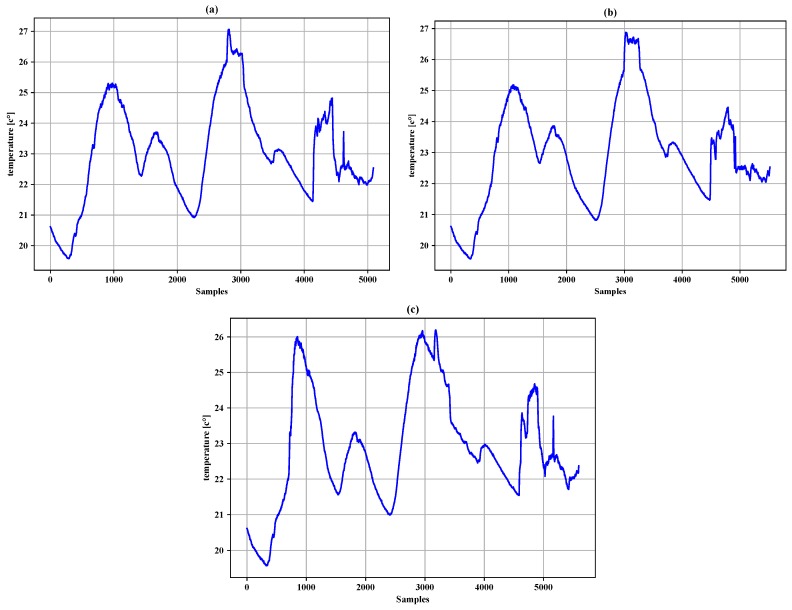
The output of the data fusion layer (**a**) the proposed approach, (**b**) LMS and (**c**) Papageorgius’s method (CLONE) with $e_{\max}$ = 0.01 for sensors 1, 2 and 3

**Figure 13 sensors-19-00795-f013:**
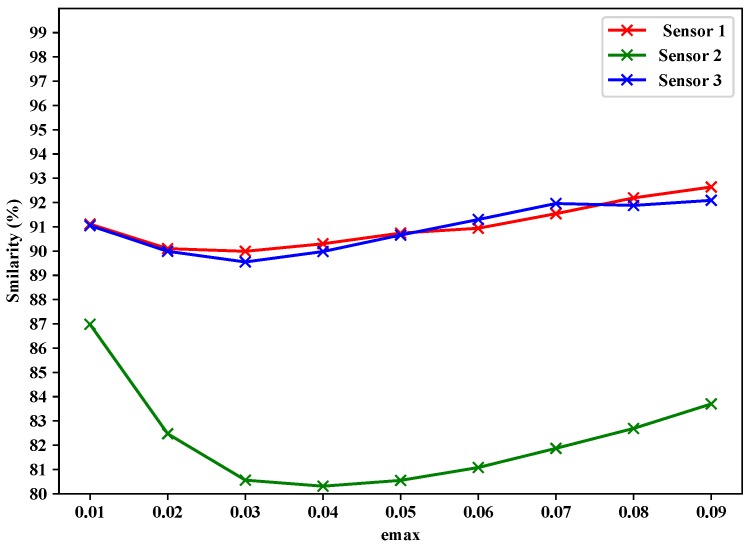
Similarity between original readings and fused reading vs. $e_{\max}$ for Sensors 1–3.

**Table 1 sensors-19-00795-t001:** The summery of parameters of Algorithm 1.

Parameter	Definition
cached_z	cache for Sensor reading when data change detected
$e_{\max}$	Maximum absolute deviation of observation from its estimated value
z(t)	Data stream produced by a sensor at time
x(t)	state value at time
e(t)	error between the observation and the estimated value
fd(t)	filtered data at time *t*
*t*	Time index = 1,2,…,*t*

**Table 2 sensors-19-00795-t002:** The summery of parameters of Algorithm 2.

Parameter	Definition
$fd^{1},fd^{2},...,fd^{n}$	The data stream of sensors passed from the data filtering layer at time *t*
$R^{1},R^{2},...,R^{n}$	Covariance matrices of each sensor calculated by Kalman filter at time *t*
$H^{1},H^{2},...,H^{n}$	Identity matrices of each sensor calculated by Kalman filter at time *t*
R(t)	The total covariance matrix at time *t*
z(t)	The total measurement vector at time *t*
H(t)	identity matrix of fused measurement vector z(t) at time *t*

**Table 3 sensors-19-00795-t003:** The initial estimated values for Sensors 1–3.

Sensor	Initial Value of *x*
Sensor 1	20.5078
Sensor 2	20.7724
Sensor 3	20.5666

**Table 4 sensors-19-00795-t004:** Comparison of the proposed approach, LMS and Papageorgius’s method (CLONE) for Sensor 1.

MAE	Incoming Data	Proposed Approach			LMS			Papageorgious’s Method (CLONE)		
		Forwarded Data No.	Reduction Percentage of Incoming Data	Reduction Accuracy	Forwarded Data No.	Reduction Percentage of Incoming Data	Reduction Accuracy	Forwarded Data No.	Reduction Percentage of Incoming Data	Reduction Accuracy
0.01	5896	3346	43.25%	92.61%	5129	13.01%	96.30%	5593	5.14%	94.91%
0.02	5896	2652	55.02%	91.59%	3774	35.99%	90.50%	5515	6.46%	93.59%
0.03	5896	2152	63.50%	86.67%	3049	48.29%	88.42%	5472	7.19%	92.86%
0.04	5896	1803	69.42%	85.11%	2560	56.58%	84.91%	5444	7.67%	92.38%
0.05	5896	1547	73.76%	82.77%	2273	61.45%	76.39%	5426	7.97%	91.21%
0.06	5896	1348	77.14%	79.92%	2044	65.33%	75.53%	5406	8.31%	91.07%
0.07	5896	1183	79.94%	77.12%	1842	68.76%	74.51%	5375	8.84%	90.87%
0.08	5896	1015	82.78%	73.75%	1661	71.83%	73.73%	5367	8.97%	89.33%
0.09	5896	877	85.13%	69.41%	1540	73.88%	71.92%	5355	9.18%	88.89%

**Table 5 sensors-19-00795-t005:** Comparison of the proposed approach, LMS and Papageorgius’s method (CLONE) for Sensor 2.

MAE	Incoming Data	Proposed Approach			LMS			Papageorgious’s Method (CLONE)		
		Forwarded Data No.	Reduction Percentage of Incoming Data	Reduction Accuracy	Forwarded Data No.	Reduction Percentage of Incoming Data	Reduction Accuracy	Forwarded Data No.	Reduction Percentage of Incoming Data	Reduction Accuracy
0.01	5896	3566	39.52%	93.30%	5219	11.48%	96.96%	5718	3.02%	97.03%
0.02	5896	2983	49.41%	90.16%	3896	33.92%	90.15%	5667	3.88%	96.16%
0.03	5896	2522	57.23%	78.70%	3075	47.85%	87.60%	5644	4.27%	95.77%
0.04	5896	2204	62.62%	77.32%	2505	57.51%	82.86%	5623	4.63%	95.42%
0.05	5896	1965	66.67%	75.22%	2160	63.36%	75.67%	5603	4.97%	95.09%
0.06	5896	1754	70.25%	72.89%	1919	67.45%	74.93%	5587	5.24%	94.81%
0.07	5896	1598	72.90%	70.24%	1711	70.98%	74.16%	5571	5.51%	94.53%
0.08	5896	1460	75.24%	66.66%	1578	73.24%	73.51%	5563	5.65%	94.40%
0.09	5896	1336	77.34%	62.60%	1465	75.15%	71.54%	5558	5.73%	94.32%

**Table 6 sensors-19-00795-t006:** Comparison of the proposed approach, LMS and Papageorgius’s method (CLONE) for Sensor 3.

MAE	Incoming Data	Proposed Approach			LMS			Papageorgious’s Method (CLONE)		
		Forwarded Data No.	Reduction Percentage of Incoming Data	Reduction Accuracy	Forwarded Data No.	Reduction Percentage of Incoming Data	Reduction Accuracy	Forwarded Data No.	Reduction Percentage of Incoming Data	Reduction Accuracy
0.01	5896	3420	41.99%	93.17%	5135	12.91%	96.79%	5658	4.04%	96.01%
0.02	5896	2742	53.49%	91.30%	3919	33.53%	90.33%	5589	5.21%	94.84%
0.03	5896	2235	62.09%	86.13%	3131	46.90%	88.37%	5551	5.85%	94.20%
0.04	5896	1853	68.57%	84.24%	2661	54.87%	83.91%	5501	6.70%	93.34%
0.05	5896	1559	73.56%	82.36%	2338	60.35%	77.09%	5464	7.33%	92.72%
0.06	5896	1330	77.44%	80.29%	2092	64.52%	76.36%	5436	7.80%	92.24%
0.07	5896	1161	80.31%	77.43%	1919	67.45%	75.72%	5387	8.63%	91.41%
0.08	5896	1039	82.38%	73.54%	1644	72.12%	74.90%	5369	8.94%	91.11%
0.09	5896	928	84.26%	68.56%	1567	73.42%	72.49%	5358	9.12%	90.92%

**Table 7 sensors-19-00795-t007:** Comparison between the proposed approach and LMS algorithm and Papageorgius’s method (CLONE) for Sensors 1–3.

MAE	Proposed Approach					LMS					Papageorgious’s Method (CLONE)				
	Filtered Data No.			Fused Data No.	DRP ${}^{1}$	Filtered Data No.			Fused Data No.	DRP ${}^{1}$	Filtered Data No.			Fused Data No.	DRP ${}^{1}$
	Sensor 1	Sensor 2	Sensor 3			Sensor 1	Sensor 2	Ssensor 3			Sensor 1	Sensor 2	Sensor 3		
0.01	3346	3566	3420	5372	69.63%	5129	5219	5135	5877	66.78%	5593	5718	5658	5897	66.67%
0.02	2652	2983	2742	4836	72.66%	3774	3896	3919	5444	69.23%	5515	5667	5589	5897	66.67%
0.03	2152	2522	2235	4324	75.56%	3049	3075	3131	4907	72.26%	5472	5644	5551	5897	66.67%
0.04	1803	2204	1853	3873	78.11%	2560	2505	2661	4363	75.34%	5444	5623	5501	5897	66.67%
0.05	1547	1965	1559	3495	80.24%	2273	2160	2338	4019	77.28%	5426	5603	5464	5897	66.67%
0.06	1348	1754	1330	3167	82.10%	2044	1919	2092	3700	79.09%	5406	5587	5436	5897	66.67%
0.07	1183	1598	1161	2883	83.70%	1842	1711	1919	3428	80.62%	5375	5571	5387	5897	66.67%
0.08	1015	1460	1039	2644	85.05%	1661	1578	1644	3119	82.37%	5367	5563	5369	5897	66.67%
0.09	877	1336	928	2420	86.32%	1540	1465	1567	2917	83.51%	5355	5558	5358	5897	66.67%

${}^{1}$ Data Reduction Percentage.

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
