# Peer review of "An In-Networking Double-Layered Data Reduction for Internet of Things (IoT)"

_sensors, 2019, doi:10.3390/s19040795_

Round 1
Reviewer 1 Report
Overall this paper is well written and structured. First it presents a new way of doing data reduction with two tiers and using Kalman Filter. The presented results are interesting and also show the benefit of proposed approach. There are a few points to be clarified.
- the data filtering layer is clearly explained while the data fusion layer requires some improvement. For example, the motivation to do data fusion over multiple sensors should be better explained, perhaps with some example or use cases.
- currently the output of the data fusion layer seems to be the total estimated value of all sensors in the same location. It is unclear that what will be sent out to the cloud. Is it only the aggregated value of all sensors in the same place or still you will send the estimated value of each sensor to the cloud? If this is the case, I guess you do not send the estimated value every time. There must be still some filtering at the data fusion layer, right?
- in terms of the result, it is better to have some figure to show the results with and without the data fusion layer
- "the proposed approach outperform LMS algorithm and Papageorgious’s Method (CLONE) in terms of
data reduction percentage and achieves high data recovery accuracy as reported by Figure 7". This conclusion is not well supported by Figure 7. A better explanation is required.
Author Response
Thanks for the reviewer’s careful examination. We have provided the replies in the attached file.

Reviewer 2 Report
The paper presents a data fusion and reduction strategy for IOT. The methodology is based on the use of Kalman filter at different stages.
The paper is based on simulated datasets alone.
The results presented seem to be of good quality but still some concerns need to be addressed
before the paper can be published.
1) Most importantly the section 3 needs to be revised considerably. The methodology is not entirely clear. A flowchart in addition to the pseudo-code might help the understanding of the information flow better.
2) The reviewer is curious about the data sets for the 3 sensors that are considered. They need to be provided and the trends shown in the data sets need to be explained
3) How can the faulty sensors be identified by the system, and in case they are identified how cane their effect on the mean measurement at the fusion layer be minimized? A possible way is through having an updating R matrix or check for outliers in the data. Few comments about this issue will improve the paper
4) Can the multi-rate date be tackled by the methodology? if it can some clarity in the writing is needed to make it obvious, if not a solution needs to be proposed.
5) A discussion on the values of A, P and R values needs to be provided. A good example of such a discussion is by Soman et al. Also the sensitivity of the method to the choice of these values needs to be provided.
Soman R., Malinowski, P., Majewska, K., Mieloszyk, M., Ostachowicz, W.: Kalman Filter based
Neutral Axis tracking in composites under changing operating conditions. Mechanical Systems
and Signal Processing, 110 (2018),485-498
The paper is also a good example for the multi-rate data fusion.
6) The manuscript claims 'Moreover, the
proposed approach attains acceptable data reduction accuracy.' This needs to be supported by appropriate references which shows that the achieved accuracy is sufficient.
7) More discussion on the tables 4,5,6 needs to be provided. IT is observed that for lower MAE, the performance of the data reduction for new method is significantly better than other methods, what could be the reason for the same?
Apart from these comments, the figures need to be improved for better visibility
Figure 2 needs to be larger
Figure 5, Figure 6 need larger font size for axis names.
There are several typographical and grammatical mistakes which need to be addressed.
The numbering for section 3.1.1 is wrong.
Author Response

(The authors gave the same response as above.)

Round 2
Reviewer 2 Report
This draft of the manuscript is considerably improved from the previous draft. But the reviewer is not satisfied with some of the discussions and the reasons provided by the authors
Firstly, for multi-rate fusion, the reviewer thinks that parallel implementation of the Kalman Filter is not efficient computationally. The incredible power of KF comes from its ability to fuse data from different sources and needs to be used.
Secondly another observations is that is the Figures 5,6,7 there appears to be a lag in the filtered data and the measurements. This lag is the highest for the proposed method. In some applications where this information needs to be used for further analysis this lag may not be acceptable. So a discussion highlighting this aspect of the performance needs to be provided. This should also be complemented with the sensitivity of this lag to different quantities including the threshold error etc.
The state matrix x has not been clearly stated, so outright definition of the x matrix is necessary
The authors have provided the P, R matrices, but the Q matrix has not been provided.
There is some text in Algorithm 2 which seems to be misplaced.
There are several typographical mistakes as well as grammar polishing is necessary in the paper.
A better arrangement of the subfigures in Figures 5,6,7 may make the paper easier to read.
Author Response
Thanks for the review’s careful examination. We have provided the reply in the attached file.
